# Health Literacy among Older Adults in Portugal and Associated Sociodemographic, Health and Healthcare-Related Factors

**DOI:** 10.3390/ijerph20054172

**Published:** 2023-02-25

**Authors:** Andreia Costa, Rodrigo Feteira-Santos, Violeta Alarcão, Adriana Henriques, Teresa Madeira, Ana Virgolino, Miguel Arriaga, Paulo J. Nogueira

**Affiliations:** 1Instituto de Saúde Ambiental, Faculdade de Medicina, Universidade de Lisboa, 1649-028 Lisboa, Portugal; 2CIDNUR—Centro de Investigação, Inovação e Desenvolvimento em Enfermagem de Lisboa, Escola Superior de Enfermagem de Lisboa, 1600-190 Lisboa, Portugal; 3CRC-W—Católica Research Centre for Psychological, Family and Social Wellbeing, Universidade Católica Portuguesa, 1649-023 Lisboa, Portugal; 4Laboratório Associado TERRA, Faculdade de Medicina, Universidade de Lisboa, 1649-028 Lisboa, Portugal; 5EPI Task-Force FMUL, Faculdade de Medicina, Universidade de Lisboa, 1649-028 Lisboa, Portugal; 6Área Disciplinar Autónoma de Bioestatística (Laboratório de Biomatemática), Faculdade de Medicina, Universidade de Lisboa, 1649-028 Lisboa, Portugal; 7Centro de Investigação e Estudos de Sociologia (CIES-Iscte), Instituto Universitário de Lisboa (Iscte), 1649-026 Lisboa, Portugal; 8Laboratório de Nutrição, Faculdade de Medicina, Universidade de Lisboa, 1649-028 Lisboa, Portugal; 9Centro de Investigação em Saúde Pública, Escola Nacional de Saúde Pública, Universidade NOVA de Lisboa, 1600-560 Lisboa, Portugal; 10Comprehensive Health Research Center, Escola Nacional de Saúde Pública, Universidade NOVA de Lisboa, 1600-560 Lisboa, Portugal

**Keywords:** health literacy, aging, healthcare, health promotion

## Abstract

Although the health literacy level of the general population was described recently, little is known about its specific levels among older adults in Portugal. Therefore, this cross-sectional study aimed to investigate the levels of health literacy demonstrated by older adults in Portugal and explore associated factors. Using a randomly generated list of telephone numbers, adults aged 65 years or more living in mainland Portugal were contacted in September and October 2022. Sociodemographic, health and healthcare-related variables were collected, and the 12-item version of the European Health Literacy Survey Project 2019–2021 was used to measure health literacy. Then, binary logistic regression models were used to investigate factors associated with limited general health literacy. In total, 613 participants were surveyed. The mean level of general health literacy was (59.15 ± 13.05; n = 563), whereas health promotion (65.82 ± 13.19; n = 568) and appraising health information (65.16 ± 13.26; n = 517) were the highest scores in the health literacy domain and the dimension of health information processing, respectively. Overall, 80.6% of respondents revealed limited general health literacy, which was positively associated with living in a difficult household financial situation (4.17; 95% Confidence Interval (CI): 1.64–10.57), perceiving one’s own health status as poorer (7.12; 95% CI: 2.02–25.09), and having a fair opinion about a recent interaction with primary healthcare services (2.75; 95% CI: 1.46–5.19). The proportion of older adults with limited general health literacy in Portugal is significant. This result should be considered to inform health planning according to the health literacy gap of older adults in Portugal.

## 1. Introduction

During past decades, countries in large parts of the world noticed the demographic aging of their population, with a growing number and share of older people (those aged 65 years or more) [1]. Different factors are indicated as reasons for this shift in population, especially those causing an increase in longevity, such as advances in medical technologies, better access to healthcare and quality and better living conditions [1,2]. However, with longer life and the biological changes related to aging [3], this shift in the population’s demographics poses a potential burden for the healthcare system with increased expenditures due to increased demand if the added years are lived in poor health [2,4].

Some experts have pointed out that the onset of diseases and shortening of the years lived with disability is being delayed in some countries [5], potentially reflecting some differences at the sociodemographic level or benefits from health policy planning as well as the effectiveness of disease prevention or health promotion interventions [6]. Patients, especially older adults, should generally be able to make appropriate choices regarding their health status, lifestyle or self-management in the case of any disease or condition [7]. Nevertheless, healthcare systems are organised in such a way, with distinct and differentiated levels of care or different responses for specific conditions, that a set of several skills is often needed for patients to manage their own or their relatives’ health adequately and navigate throughout the system [8]. In this context, the interest in studying health literacy has grown, and its relevance in informing health policies, as well as its improvement [9], is currently seen as a priority to empower people of different age groups [8,10], either at a regional or country-wide level [11]. The recognition of the need to measure health literacy in Europe led to the establishment of the European Health Literacy Project (HLS-EU), which developed instruments to assess and compare health literacy across different European countries [12], and the evolution of the project over time [13].

Although a multiplicity of definitions around the concept of health literacy can be found in the literature, the one suggested by Sørensen and colleagues was considered for this study, i.e., that it involves “people’s knowledge, motivation and competencies to access, understand, appraise, and apply health information in order to make judgments and take decisions in everyday life concerning healthcare, disease prevention and health promotion to maintain or improve quality of life during the life course” [8]. According to the evidence, the literature has described an association between poorer health outcomes and the use of health care services with a lower level of health literacy [14]. Moreover, an inadequate level of health literacy was described among older adults [15,16], a population subgroup with a higher risk of disease and healthcare use. In the context of the COVID-19 pandemic, it was also suggested that higher levels of health literacy in older people can be either a protective factor against depression or promote healthier behaviours [17]. Still, a social gradient associated with health literacy was observed in the HLS-EU not only when considering age groups, but also for other sociodemographic characteristics of the population, such as educational level or financial deprivation, in conjunction with healthcare characteristics [9,13,18,19].

The health literacy of the general population was measured in Portugal during the HLS-EU in 2021 [20]. However, the evidence regarding the levels of health literacy and related measures for older adults is still limited, as well as regarding the factors associated with limited health literacy. Therefore, the main objective of this study was to describe the levels of general and health literacy-related measures among a representative sample of the population of 65 years or more living in mainland Portugal. Moreover, this study also aimed to investigate the sociodemographic, health status and healthcare-related factors associated with limited general health literacy among older adults in Portugal.

## 2. Materials and Methods

Data from the “Knowing Social Prescribing needs of the elderly” (PROKnos) study were used for this work. The PROKnos project followed a mixed-method approach with quantitative (survey) and qualitative (focus groups) components, aiming to understand older people’s needs on social prescribing to promote healthy and active aging. Besides describing the perception of the older regarding social prescribing, PROKnos also characterised the quality of life, well-being and general health literacy of people aged 65 years and older.

The PROKnos project was submitted, appreciated and obtained ethical approval by the Ethics Committee of the Centro Académico de Medicina de Lisboa (process number 193/22). Furthermore, all procedures followed the Code of Ethics of the Declaration of Helsinki for observational studies [21].

### 2.1. Study Design and Sampling

The survey followed a cross-sectional design and involved a random representative sample of people aged 65 years or over living in mainland Portugal. A minimum sample size of 384 individuals was calculated using the OpenEpi tool [22], ensuring a 95% confidence interval for proportions calculated and achieving a desired absolute precision of 5%.

People were contacted by phone using a list of fixed and mobile telephone numbers randomly generated by a specialised centre in polling and public opinion research. If a given telephone number was active, then a maximum of three unsuccessful contact attempts were made before excluding it. Telephone numbers registered to public or private organisations were also excluded from this study. When contacted, subjects were eligible to enrol in the study if they (1) were living in mainland Portugal, (2) were at least 65 years old and (3) accepted to participate in the study after being adequately informed about its purpose, involved entities and data collection procedures.

At least one contact attempt was made to 1916 randomly generated telephone numbers. From those, establishing contact was possible with 1757 subjects, whereas 159 did not answer any attempted calls. Among those contacted, 924 were excluded because they did not fulfil the inclusion criteria, and 220 refused to participate or decided to leave the study during the interview. The answer rate was 73.6%, which was calculated using the number of participants (n = 613) divided by the sum of the number of participants and the number of successful contacts to people fulfilling the inclusion criteria but not completing the survey (n = 833).

### 2.2. Data Collection

This survey was promoted by a research consortium including the Instituto de Saúde Ambiental, Universidade de Lisboa and Escola Superior de Enfermagem de Lisboa, and it was conducted by the Centro de Estudos e Sondagens de Opinião, Universidade Católica Portuguesa. Data collection was carried out between the 20th of September and the 3rd of October 2022. Structured telephone interviews were performed by a trained team of 44 interviewers who used a Computer-Assisted Telephone Interviewing (CATI) system. Each interview lasted about 37 min on average. Individuals were invited to participate after being informed about the purpose of the study and were only enrolled if verbal consent was given to the interviewer.

### 2.3. Measures

The sociodemographic characteristics of the sample were collected at the beginning of the telephone interview. Collected variables included age, which was categorised into age groups (65–74 years; 75+ years), gender (male; female), household living arrangement (living alone; living with others), country of birth (born in Portugal; born outside Portugal), place of residence (based on the nomenclature of territorial units for statistics, level II (NUTS II)) (North; Centre; Lisbon Metropolitan Area; Alentejo; Algarve), educational level (up to 2nd cycle of primary education; 3rd cycle of primary education; high school; university education); employment status (working professionally; not working professionally); self-perceived household financial situation (comfortable or very comfortable; sufficient; difficult or very difficult). Health status and healthcare-related variables were also collected, namely self-perceived health status (good or very good; fair; bad or very bad); self-reported chronic disease or disability (yes; no); enrolment in primary healthcare (yes; no); having had a primary healthcare medical consultation at least once in the last six months (yes; no); self-rated primary healthcare evaluation for the last six months (bad or very bad; fair; good or very good).

The general health literacy level was assessed using the HLS_19_-Q12, developed within the “HLS_19_—the International Health Literacy Population Survey 2019–2021” of M-POHL. This instrument is a short-form scale validated across 17 countries resulting from the HLS_19_-Q47, a 47-item instrument adapted from the original instruments used in the European Health Literacy Survey. The Portuguese version was studied and validated, involving a population sample of 16 years and older [20]. This 12-item version measures three health literacy domains and four dimensions of health information processing. The domains are health care, disease prevention and health promotion. The dimensions of health information processing are to access/obtain information relevant to health, to understand information relevant to health, to appraise/judge/evaluate information relevant to health and to apply/use information relevant to health. Participants rated each question using a bipolar 4-point Likert-type scale with the options “1 = very difficult”, “2 = difficult”, “3 = easy” and “4 = very easy”. A general health literacy score can be calculated as the sum of the numeric values from each valid item response and then scaled to a range from 0 to 100 [23], with higher values being indicative of higher levels of health literacy. This tool allows the calculation of seven sub-indices of health literacy, one for each of the three health literacy domains and the four dimensions of health information processing, using the same method of summing the values and converting them to a 0–100 measure. Appendix A shows the instrument questions and the items used to calculate each of the seven sub-indexes for health literacy domains and health information processing dimensions. A minimum of 80% of the items used to calculate each health literacy score must contain valid responses. The score was set to “missing” if this request was not verified. Then, the general health literacy score and the seven sub-indexes were categorically transformed to express four levels of health literacy, i.e., “excellent”, “sufficient”, “problematic” and “inadequate”. The procedure transformed the 0–100 scores according to the following criteria: “inadequate”, from 0 to 50 (incl.); “problematic”, from 50 to 66.67 (incl.); “sufficient”, from 66.67 to 83.33 (incl.); “excellent”, above 83.33. Another variable was also computed by combining “problematic” and “inadequate” levels into “limited health literacy”. Although two methods for health literacy score categorisation were initially established by the research team who developed and validated the HLS_19_-Q12, the one employed here was selected, as it is preferably endorsed [23]. The use of the HLS_19_ instrument in this research was only possible after permission was granted by the HLS_19_ Consortium (link: https://m-pohl.net/Design_Methods, accessed on 8 June 2022).

### 2.4. Statistical Analysis

IBM SPSS Statistics^®^ for Windows (version 26.0, 2019, IBM Corp, Armonk, NY, USA) was used for performing the statistical analyses, and the statistical significance was set to 5%. A descriptive analysis of the sociodemographic characteristics of the sample was performed as well as analyses for variables related to health status and healthcare utilisation. The analysis included the description of the level of general health literacy and the level of each sub-index of health literacy by their means and standard deviations. Moreover, the relative distribution of each level for the four health literacy levels and dichotomised variables are presented as percentages calculated based on the number of valid values. The number of total respondents whose responses were used for each score calculation and the number of missing values are also reported. 

Bivariate analyses were performed using Fisher’s exact and chi-square tests to evaluate potential associations between health literacy measures and the sociodemographic, health and healthcare-related variables.

General health literacy, as categorised as “limited” or “not limited”, was then used in a binary logistic regression model as the dependent variable. If bivariate analysis revealed a *p*-value < 0.100 to predict limited health literacy among older Portuguese adults, then sociodemographic, health status and healthcare-related variables were used as independent factors. The forward: LR method was used in the first approach model. An enter method bloc approach was then adopted to include variables of interest not contained in the first approach’s model. If any of the variables representing age, sex, education level, place of residence, or household financial situation were not included in the model, then the missing variables were entered into the final model through a second bloc approach. Results are presented as crude (cOR) and adjusted odds ratios (aOR) with 95% confidence intervals (95% CI).

Residual probabilities resulting from the final adjusted binary logistic regression model were saved and used to calculate the area under the Receiver Operating Characteristic (ROC) curve.

## 3. Results

### 3.1. Sample Characterisation

In total, 613 valid interviews were performed. In this sample, 52.2% were men, and the mean age was 72.84 years (standard deviation: 5.79 years), whereas the median age was 72 years (interquartile range: 68–76). The minimum age was 65, and the maximum age was 93. The detailed results are presented in Table 1.

Almost half of the participants were residing in the Lisbon Metropolitan Area (Lisbon MA) and comprised 40.1% of the participant pool, followed by the North region, with 26.8% of participants, and the Centre region, with 21.9%. Most participants were born in Portugal (94.3%). More than one-third of the surveyed sample had an educational level below the third cycle of primary education (34.3%), whereas 30.8% had some university education. About 70% cohabited with at least one other person, and almost 94% did not work professionally, with the majority being retired from active work life (88.4%, results not presented). Regarding perceived household financial situations, 41.6% reported a financial status adequate for their needs, and although about 28% reported a comfortable or very comfortable situation, 29.4% reported a difficult or very difficult situation.

Participants were also asked about their self-perceived health status. Almost half of the participants reported having fair health (49.8%), followed by those who perceived good or very good health status (31.6%) and by those who considered their health to be bad or very bad (18.6%). About 47% reported having at least one chronic disease or health condition.

Additionally, participants were also asked about some healthcare-related aspects. The majority indicated that they were registered for health assistance in a primary healthcare centre (95.3%), and 68% of those had had at least one medical appointment with a primary healthcare service within the previous six months. Lastly, participants were asked to rate their interaction with primary healthcare services in the previous six months, and they mainly rated them as good or very good (46.4%). More than 31% rated their previous 6 months of experience in primary healthcare as fair, and almost 13% rated it as bad or very bad.

### 3.2. Health Literacy Measures

Figure 1 shows a summary of health literacy measures, namely the general health literacy score and the seven sub-scores of health literacy as well as the percentages observed for each level of health literacy. Each measure was calculated based on the information of a range of participants from 390 to 568, depending on how many provided at least 80% of valid responses in items used to calculate each score. 

A mean score of 59.15 (±13.05) for general health literacy was observed. This value was lower than each of the means observed in each of the seven sub-indexes, which ranged between 59.53 (16.77) for the “Access” dimension of processing health information and 65.82 (13.19) for the “Health promotion” domain of health literacy.

Regarding the four levels of health literacy categorisation, the results show a higher proportion of participants in the “Problematic” and the “Inadequate” categories for some health literacy measures. The highest share of participants with a “Problematic” level of health literacy was observed in the “Apply” dimension of health information processing (72.1%), and the lowest was in the general health literacy categoric variable (55.2%). The percentages in the “Inadequate” level ranged from 9.1% to 26.7% in the “Appraise” and the “Access” dimensions of processing health information, respectively. The level of general health literacy was classified as “Sufficient” for 15.6% of participants, whereas the lowest level was observed in the “Understand” dimension (8.3%). The “Excellent” level scored the lowest values among all categories, ranging from 3% in “Healthcare” related health literacy to 8% in the “Appraise” dimension of processing health information.

### 3.3. Limited Health Literacy Distribution within Sample Subgroups

Table 2 presents the results of considering together the “Inadequate” and “Problematic” categories of general health literacy in a new measure called “Limited health literacy”. Among participants with at least 80% of valid responses in the 12 items of the HLS_19_-Q12 tool (n = 563), 80.6% presented limited health literacy.

The highest proportions of limited health literacy were observed for participants who were 75 years or older (85.5%), men (81%), living alone (83.5%), born in Portugal (81.4%), residing in the Centre region (87.6%), not working professionally (81.6%), had an educational level up to the second cycle of primary education (91.7%) and perceived their household financial situation as difficult or very difficult (95.0%).

Regarding variables related to participants’ health statuses and healthcare experiences, the categories with the highest percentages of limited health literacy were those participants who rated their health status as bad or very bad (96.3%), reported having any chronic disease or health condition (86.2%), were registered for primary healthcare assistance (80.9%), reported at least one appointment in primary healthcare within the previous six months (83.5%) and rated their recent interactions with primary healthcare as fair (90.1%).

Significant associations between the sociodemographic and health-related characteristics and limitations in health literacy were found for age groups, employment status, education level, perceived household financial situation, self-perceived health status, self-reported chronic diseases or health conditions, primary healthcare appointments in the previous six months and evaluation of primary healthcare interactions.

### 3.4. Determinants of Limited Health Literacy

The variables with statistical differences in limited health literacy between categories were the same, which satisfied the criterion for being selected for regression analysis (*p* < 0.100). Table 3 presents the crude and adjusted ORs of limited health literacy according to significant sociodemographic, health and healthcare-related characteristics obtained in binary logistic regression analyses.

Values for chances of limited health literacy ranged from an increase of 66% for those who had had a primary care appointment in the previous six months to an increase of almost 12-fold for those who perceived their own health status as bad or very bad.

In the first regression analysis approach (adjusted model, first bloc), participants aged 75 years or more who perceived a household financial situation as enough for their needs or as difficult or very difficult, those who perceived their health status as bad or very bad and those who rated the experience with primary healthcare services in the previous six months as fair had higher odds of limited health literacy. A second approach (adjusted model, second bloc) was performed, adjusting ORs for some core variables initially dropped from the final model in the first approach using the forward method, namely educational level, whereas two others had not satisfied the criterium in the univariate regression analyses (sex and place of residence). After introducing these three variables, age and perceived household financial situation lost statistical significance. In the final adjusted logistic regression model, those who perceived health status as bad or very bad registered sevenfold higher odds of demonstrating limited health literacy (7.12; 95% CI: 2.02–25.09) compared to those who perceived it as good or very good. In addition, those who perceived their household financial status as difficult or very difficult had a fourfold higher chance of presenting limited health literacy (4.17; 95% CI: 1.64–10.57) compared to those who perceived their situation as comfortable or very comfortable. Furthermore, an evaluation of the previous six months of experience with primary healthcare services as fair was associated with an increase in the odds of having limited health literacy almost threefold (2.75; 95% CI: 1.46–5.19) compared to the levels of literacy registered among those who rated it as good or very good.

The area under the ROC curve was calculated using the final adjusted model and was 78.7%.

## 4. Discussion

In this study, the levels of general and health literacy-related measures found in a representative sample of the population 65 years or more living in mainland Portugal were described. The association between limited general health literacy and sociodemographic, health and healthcare-related factors was also investigated. Studying the level of health literacy of a population is gaining importance. In Europe, it was assessed and is supposed to be periodically monitored in the future by the HLS-EU. The purpose that leads countries to assess health literacy is to know the needs of general and vulnerable population subgroups and their potential difficulties when interacting with healthcare services while also providing information to design and plan adequate health interventions [8,12,24]. Moreover, an inverse association was suggested between better health literacy and the reduction of health inequality [25], a topic that is even more relevant in an aging population. Altogether, it is considered crucial to monitor health literacy when considering health and well-being.

Our results show that about 80% of the surveyed sample demonstrated a limited level of general health literacy (65–74 years old: 78%; 75+ years old: 85%). This proportion of persons with a level of general health literacy categorised as “Inadequate” or “Problematic” in older adults in Portugal seems to be higher than that previously reported both in that country (66–75 years old: 65.5%; 76+ years old: 81.7%) [26] and as found on average in other European countries, as shown by the HLS-EU in 2015. Then, the HLS-EU reported limited health literacy percentages of 58.2% among those aged 65 to 75 years and 60.8% among those older than 75 years, even though a previous and more comprehensive tool for assessing health literacy was used [12]. The proportion of limited general health literacy found among older Portuguese adults is also much higher than those observed in previous research of other elderly populations [27,28] but lower compared to that found in Turkey (85%) [29]. Although a direct comparison between studies can present some limitations, these findings suggest a bigger difficulty for older Portuguese adults in using health information compared to their counterparts. As expected, and in line with other studies, the proportion of limited health literacy among the elder population is higher than that observed for the general adult Portuguese population (30%) [20]. Studies performed in other European populations also found similar contrasts between limited general health literacy percentages observed within older adults and other age groups [27,28,30,31,32]. The relationship between age and lower levels of health literacy has been referred to in the literature [14,16], though with inconsistent results [13]. It is noteworthy that older population usually follows a pattern of higher healthcare utilisation [14], which suggests that they are confronted more often with health issues, and that their health literacy can be challenged more frequently. Poorer outcomes in health interventions for those with lower levels of health literacy were previously described in the literature [33,34]. Altogether, one may hypothesise that older Portuguese adults may face disadvantages in using healthcare services and benefiting from health promotion or disease prevention programs. 

The results reveal that there are specific subgroups with worse results in terms of health literacy, eliciting the social gradient of health literacy [12,13]. These results are relevant because they highlight the most vulnerable groups in terms of health literacy among the Portuguese aged 65 years or more, allowing the segmentation and adaptation of health interventions according to health literacy at baseline. Greater and more notable differences between subgroups were observed across participants with different educational levels, household financial situations, and perceived health statuses. Likewise, a higher level of limited health literacy was described in the literature in some vulnerable populations’ subgroups. Using a similar but more comprehensive assessment tool of health literacy, i.e., the HLS-EU-47, the HLS-EU’s results concerning the general population in eight European countries revealed higher proportions of limited health literacy for people with less education and for those who perceive their health status as bad or very bad [12]. Sørensen and colleagues also noticed a poorer level of health literacy among those experiencing problems paying bills [12]. This relationship can be interpreted as a similar result to that found in this study with participants who reported difficult or very difficult household financial situations, suggesting that people in poor financial situations can also have more problems dealing with health information. These results are relevant, as the literature also describes higher patterns of disease, burden and disability within the unfavoured [35,36,37] in addition to low health literacy [14,38], which raises the question about the role of health literacy in health status [24], namely whether it is a determinant per se or if it interacts with other determinants, such as educational level and financial status.

This study’s results also show remarkable percentages of people scoring poorly for all other health literacy measures, i.e., health literacy domains and dimensions of health information processing. These difficulties in dealing with health information are common to other countries, as discussed earlier, but were not known for older Portuguese adults with this level of detail. Even though values did not vary markedly across domains and the dimensions of health information processing, the results reveal that older people in Portugal perceived the highest difficulties dealing with information regarding healthcare and disease prevention and somewhat fewer difficulties dealing with information concerning health promotion. Looking at the dimensions of information processing, participants perceived the most significant difficulties in appraising and judging information relevant to their health. Proportions of people reporting difficulties in accessing, understanding and applying health information were similar but slightly lower. Accordingly, health literacy measures with the lowest and the highest proportions of limited health literacy found in this study were the same as those recently reported in a study involving a sample of the general Portuguese population between December 2020 and January 2021 [20], even though it showed a much higher range of values.

The present work also reveals some factors associated with limited general health literacy, namely living with a household financial situation perceived as difficult or very difficult, perceiving one’s own health status as bad or very bad, and having a fair opinion about recent interactions with primary healthcare services, in the final adjusted regression model. The area under the ROC curve of the model showed an acceptable capacity to discriminate limited general health literacy [39]. Therefore, these results contribute to the discussion of the hypothesis regarding health literacy determinants, though establishing this relationship is not permitted due to the cross-sectional nature of the collected data. 

Poorer self-rated health status was associated with limited general health literacy. In fact, limited health literacy was already described as a determinant of self-assessed health and potentially of actual health status [24]. This result may be partly explained by the relationship between a low level of health literacy and an insufficient level of protective health behaviours among the elderly, e.g., insufficient physical activity or fruit and vegetable consumption, which was already reported [40].

On the other hand, one’s financial situation is probably a determinant of health literacy, as found in previous research regarding older [27] and general populations [12]. Methods used to measure financial deprivation have not always been the same across studies, though the results they provide can be interpreted as a proxy, as in the case of self-perception of a difficult household financial situation used here. Indeed, it was already pointed out that individuals from higher socioeconomic statuses can potentially have more ability and access to resources that allow them to self-manage their health status better [41], although this association is not straightforward [42].

The association between limited general health literacy and the evaluation of primary healthcare services is more difficult to interpret in light of the existing literature because the latter has not been frequently investigated. This study found that a fair evaluation of primary healthcare services was significantly associated with limited health literacy when compared to a good or very good rating, but the association with a bad or very bad evaluation was not significant in the final model, suggesting that other factors may contribute more to explain the limited health literacy of those that evaluate primary healthcare interactions as worse.

Contrary to the results found in other research studies evaluating the general population [12,31,32] or even the elderly [27,28], after adjusting for other variables of interest, neither education nor age was associated with limited general health literacy. Notably, the World Health Organization considered people with a low educational level at risk of low literacy in a 2013 report [43]. The age distribution of the included sample may have reduced the heterogeneity between participants and partly explain this result. Nevertheless, considering that one’s self-perceived financial situation is closely linked to their level of education, it is also likely that the latter is suppressed by the former in the adjusted models, resulting in a decrease of its strength.

This study used a newly collected database with data from a representative sample of the older population in Portugal. It allowed for deepening the knowledge about health literacy in older age groups by looking at the distinct health literacy domains and dimensions of health information processing. However, some limitations need to be acknowledged regarding the methodological aspects of this study. First, the HLS_19_-Q12 tool, though validated, provides a subjective measurement of health literacy, which does not allow the assessment of actual functional health literacy, i.e., how people perform when dealing with health information. Other research (65–79 years old sample) shows a slightly higher level of limited functional health literacy than the proportion found here [44]. Second, the distribution of participants across the five mainland Portugal regions deviated a little from what was observed in the last Portuguese population census [45], especially regarding the proportions of men and women and participants living in the North and the Lisbon Metropolitan Area. Moreover, it is essential to note that there is an overrepresentation in the present sample of participants who had completed at least the high school level of education. To accommodate these differences in the sample characteristics, we forced the inclusion of these sociodemographic variables in the final regression model. Lastly, the cross-sectional nature of the collected data does not allow conclusions on the causality of sociodemographic or health-related characteristics in determining a limited level of general health literacy.

Further investigation may include other variables, e.g., health morbidity and health behaviour (smoking habits, physical activity or dietary habits), to obtain a more comprehensive knowledge of risk groups and factors associated with health literacy. Finally, other approaches to data analysis can be performed in order to evaluate the role of health literacy as a determinant for specific morbidity or disability and to investigate the actual burden of low health literacy among older Portuguese adults.

## 5. Conclusions

The level of limited general health literacy among the elderly is significantly high, as about 80% revealed having difficulties in dealing with health information. This study’s findings also demonstrate that, after adjusting for other characteristics, living with a financial situation perceived as difficult or very difficult, perceiving one’s own health status as bad or very bad and having a fair opinion about interactions with primary healthcare services were factors associated with a limited level of general health literacy. Particularly, the fact that potential financial deprivation is associated with difficulties in dealing with health information puts vulnerable people in a position where inequalities can be exacerbated. The results demand a thorough approach to address the literacy gap of older Portuguese adults in dealing with health information, as they are likely to be tested more often with health issues and need to interact with healthcare services more frequently. The present findings reveal additional challenges when designing and implementing health policies and interventions for health promotion and disease prevention targeted at older persons. Health planners and decision-makers should consider these limitations in health literacy when addressing active and healthy aging.

## Figures and Tables

**Figure 1 ijerph-20-04172-f001:**
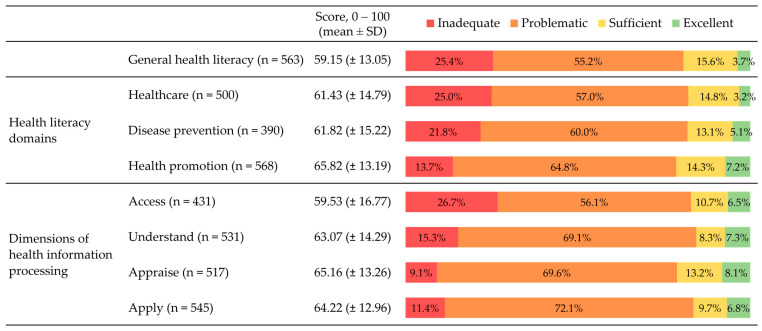
Scores (mean ± SD) and levels (%) of health literacy. The n varies across health literacy measures because it represents the number of participants with at least 80% of valid responses in the items used for calculating each score. SD, Standard deviation.

**Table 1 ijerph-20-04172-t001:** Sociodemographic, health and healthcare-related characteristics of the surveyed individuals.

Sociodemographic Characteristics	
Age (mean ± SD ^1^)	72.84 ± 5.79
Age groups	n (%)
65–74 years	405 (66.1%)
75+ years	208 (33.9%)
Gender	
Male	320 (52.2%)
Female	293 (47.8%)
Household living arrangement	
Living alone	182 (29.7%)
Living with others	431 (70.3%)
Country of birth	
Portugal	578 (94.3%)
Other country	35 (5.7%)
Place of residence (based on nomenclature of territorial units for statistics, level II region)	
North	164 (26.8%)
Centre	134 (21.9%)
Lisbon Metropolitan Area	246 (40.1%)
Alentejo	37 (6.0%)
Algarve	32 (5.2%)
Employment status	
Working professionally	38 (6.2%)
Not working professionally	574 (93.6%)
Do not know/Did not answer	1 (0.2%)
Educational level	
Up to 2nd cycle of primary education	210 (34.3%)
3rd cycle of primary education	83 (13.5%)
High school	131 (21.4%)
University education	189 (30.8%)
Perceived household financial situation	
Comfortable or very comfortable	173 (28.2%)
Enough for needs	255 (41.6%)
Difficult or very difficult	180 (29.4%)
Do not know/Did not answer	5 (0.8%)
Health status and healthcare-related variables	
Self-perceived health status	
Good or very good	194 (31.6%)
Fair	305 (49.8%)
Bad or very bad	114 (18.6%)
Self-reported chronic disease or disability	
Yes	287 (46.8%)
No	326 (53.2%)
Registered in a primary healthcare centre	
Yes	584 (95.3%)
No	29 (4.7%)
Primary healthcare medical consultation within the previous six months, at least once (n = 584)	
No	186 (31.8%)
Yes	398 (68.2%)
Primary healthcare evaluation, previous six months (n = 584)	
Bad or very bad	74 (12.7%)
Fair	184 (31.5%)
Good or very good	271 (46.4%)
Do not know/Did not answer	55 (9.4%)

^1^ SD, Standard deviation.

**Table 2 ijerph-20-04172-t002:** General health literacy score means and limited health literacy by sociodemographic, health and healthcare-related characteristics.

	General HL Mean (±SD)	Limited HL n (%)	*p*-Value ^1^
Total (n = 563)	59.15 (±13.05)	454 (80.6%)	
Sociodemographic characteristics			
Age groups			
65–74 years	59.79 (±13.64)	295 (78.2%)	0.042
75+ years	57.84 (±11.70)	159 (85.5%)	
Gender			
Male	59.87 (±12.68)	239 (81.0%)	0.831
Female	58.35 (±13.43)	215 (80.2%)	
Household living arrangement			
Living alone	58.21 (±13.71)	137 (83.5%)	0.292
Living with others	59.53 (±12.77)	317 (79.4%)	
Country of birth			
Portugal	58.83 (±12.91)	432 (81.4%)	0.103
Other country	64.50 (±14.40)	22 (68.8%)	
Place of residence (based on NUTS II region)			
North	59.41 (±13.81)	116 (77.9%)	
Centre	56.13 (±12.47)	106 (87.6%)	
Lisbon Metropolitan Area	60.41 (±13.12)	182 (78.8%)	0.174
Alentejo	60.68 (±12.58)	24 (75.0%)	
Algarve	58.61 (±9.93)	26 (86.7%)	
Employment status			
Working professionally	65.05 (±11.89)	24 (66.7%)	0.047
Not working professionally	58.76 (±13.05)	429 (81.6%)	
Educational level			
Up to 2nd cycle of primary education	53.16 (±12.16)	165 (91.7%)	
3rd cycle of primary education	58.56 (±11.94)	64 (86.5%)	<0.000
High school	59.06 (±11.85)	104 (81.9%)	
University education	65.37 (±12.37)	121 (66.5%)	
Perceived household financial situation			
Comfortable or very comfortable	65.69 (±12.74)	106 (64.2%)	
Enough for needs	59.74 (±11.34)	193 (82.1%)	<0.000
Difficult or very difficult	51.56 (±11.91)	152 (95.0%)	
Health status and healthcare-related variables			
Self-perceived health status			
Good or very good	65.14 (±12.41)	125 (68.7%)	
Fair	58.52 (±11.61)	226 (82.5%)	<0.000
Bad or very bad	50.57 (±12.51)	103 (96.3%)	
Self-reported chronic disease or disability			
Yes	56.59 (±12.73)	231 (86.2%)	0.002
No	61.47 (±12.93)	223 (75.6%)	
Registered in primary healthcare			
Yes	58.97 (±13.07)	432 (80.9%)	0.474
No	62.36 (±12.48)	22 (75.9%)	
Primary healthcare medical consultation last six months, at least once (n = 534)			
No	61.63 (±13.22)	128 (75.3%)	0.033
Yes	57.73 (±12.83)	304 (83.5%)	
Primary healthcare evaluation, last six months (n = 487)			
Bad or very bad	54.29 (±13.43)	60 (88.2%)	
Fair	55.20 (±12.48)	146 (90.1%)	0.001
Good or very good	61.57 (±12.16)	196 (76.3%)	

Footnotes: ^1^ Fisher’s exact or chi-square tests used to evaluate associations between limited health literacy and sociodemographic, health and healthcare-related variables. HL, health literacy; SD, standard deviation.

**Table 3 ijerph-20-04172-t003:** Multivariate logistic regression analysis to predict limited general health literacy.

	Crude OR (95% CI) ^1^	Adjusted OR, 1st bloc (95% CI) ^2^	Adjusted OR, 2nd bloc (95% CI) ^3^
Age groups			
65–74 years	1	1	1
75+ years	1.64 (1.02–2.63)	1.89 (1.06–3.35)	1.78 (0.99–3.23)
Employment status			
Working professionally	1	-	-
Not working professionally	2.21 (1.07–4.58)	-	-
Educational level			
University education	1	-	1
High school	2.28 (1.32–3.94)	-	1.91 (0.98–3.74)
3rd cycle of primary education	3.23 (1.55–6.72)	-	1.86 (0.75–4.65)
Up to 2nd cycle of primary education	5.55 (3.01–10.22)	-	1.84 (0.87–3.90)
Perceived household financial situation			
Comfortable or very comfortable	1	1	1
Enough for needs	2.56 (1.61–4.06)	1.85 (1.07–3.20)	1.78 (0.995–3.17)
Difficult or very difficult	10.58 (4.85–23.05)	4.95 (2.11–11.61)	4.17 (1.64–10.57)
Self-perceived health status			
Good or very good	1	1	1
Fair	2.15 (1.38–3.34)	1.56 (0.92–2.64)	1.41 (0.80–2.48)
Bad or very bad	11.74 (4.12–33.45)	8.29 (2.40–28.69)	7.12 (2.02–25.09)
Self-reported chronic disease or disability			
Yes	1	-	-
No	2.02 (1.30–3.12)	-	-
Primary healthcare medical consultation last 6 months, at least once (n = 534)			
No	1	-	-
Yes	1.66 (1.07–2.60)	-	-
Primary healthcare evaluation, last six months (n = 487)			
Good or very good	1	1	1
Fair	2.84 (1.57–5.13)	2.64 (1.42–4.91)	2.75 (1.46–5.19)
Bad or very bad	2.33 (1.06–5.15)	1.68 (0.73–3.90)	1.74 (0.74–4.09)
Gender			
Male	-	-	1
Female	-	-	0.87 (0.51–1.47)
Place of residence (based on NUTS II region)			
North	-	-	1
Centre	-	-	1.74 (0.81–3.76)
Lisbon Metropolitan Area	-	-	1.27 (0.67–2.40)
Alentejo	-	-	1.05 (0.36–3.12)
Algarve	-	-	4.41 (0.87–22.49)

OR, odds ratio; CI, confidence interval. ^1^ Binary logistic regression model (univariate analyses, not adjusted). ^2^ Binary logistic regression model (forward, LR method) adjusted for age groups, educational level, employment status, perceived household financial situation, self-perceived health status, self-reported chronic disease or health condition, medical consultation in primary healthcare within the previous six months and rating of the previous six months of experience with primary healthcare services. ^3^ Binary logistic regression model (1st bloc: forward, LR method; 2nd bloc: enter method) adjusted for sex, age groups, educational level, place of residence, employment status, perceived household financial situation, self-perceived health status, self-reported chronic disease or health condition, medical consultation in primary healthcare within the previous six months and rating of the previous six months of experience with primary healthcare services.

## Data Availability

The datasets used and/or analysed during the current study are available from the corresponding author upon reasonable request.

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
