# Peer review of "Health Literacy among Older Adults in Portugal and Associated Sociodemographic, Health and Healthcare-Related Factors"

_ijerph, 2023, doi:10.3390/ijerph20054172_

Round 1

Reviewer 1 Report

This manuscript could be an important reference for future studies. However, is still needed to improve the quality of this paper. Please revise the manuscript to address the expressed concerns. After thorough review, I am recommending some revisions. In this regard, kindly address the following comments and suggestions to further improve your manuscript

a.       Please write the type of study, sample size, sampling strategy and date and country of study in abstract

b.       It was better if you wrote some of main finding as quantitative or mean ±SD within the abstract. The result section in the abstract is poor and immature!!

c.       The introduction section need some revision. You could summarize this section a bit more for readers. Write about the problems, the novelty of your study, and your study goals within the introduction. In this section, you can use the following articles:

1-      Examining Health literacy levels and its Association with Demographic Dynamics among Intra-City Commercial Drivers: Results from a Survey in Nigeria

2-      Evaluation of Health Literacy in Academics at a University of Turkey

d.       The methods need to be improved by providing more detail information related to participant’s selection (e.g. respond rate; necessary permissions from who? How did the researcher contact the potential participants?)

e.       What was your sample size formula? What is your expected power? please mention in main text

f.        Discuss more about your sampling strategy? The structure of your sampling is so vague and understandable. Did you have sampling frame? how did you access to this frame

g.       write about all applied exclusion and inclusion criteria a bit more clearly by which you selected samples for this survey.

h.       You could increase the number of more recently studies in the reference section. You should have comprehensive and reliable comparisons between your findings with the other previous studies. Furthermore, write about the limitations of your survey. Are there any limitations for this study? If yes, please mention all limitations of current study within the discussion section, too.

Author Response

please find reply in attached file

Reviewer 2 Report

Well-designed research and well-written.

It is of course a known fact that statistic correlations do not always guarantee causality; and that there are limits to the reliability of self-reported data. Yet readers will be aware of that and you present your conclusions cautiously.

A very different approach would be to let people answer content-related questions after reading a brochure, for example. This was outside the scope of your research.

Attached: your text with a number of remarks about details.

Author Response

please find reply in attached file

Round 2

Reviewer 1 Report

thank you for the revised manuscript and detailed responses to my previous suggestions. I find the revised manuscript to be much more clear and more comprehensible. Because you addressed my previous suggestions, I find the manuscript ready to be published